# Vibrational Enthalpies of Solid Crystalline Materials

**Christopher Martin Stanley** (ID)

R.B. Annis School of Engineering, University of Indianapolis, Indianapolis, IN 46227, USA; stanleyc@uindy.edu

**Abstract:** Specific heat at constant pressure is traditionally a difficult thermodynamic quantity to obtain from first-principles calculations. While theoretical avenues to $C_p(T)$ do exist—most notably, the quasi-harmonic approximation—there are many materials for which this approximation is not valid. One of those materials is Ge. In this paper, we demonstrate how a new method—termed the Beyond Quasi-Harmonic method—takes into account all anharmonic vibrations by showing how our results are significantly better than those achieved through using the quasi-harmonic model. In addition, we calculate $C_p(T)$ for 3C-SiC, a material for which there are surprisingly few experimental results. For 3C-SiC, our results agree well with the available experiments, and for Ge, our results agree very well with the generally accepted values.

**Keywords:** Beyond Quasi-Harmonic method; specific heat; heat capacity; supercell preparation; vibrational enthalpy; vibrational energy; phonon; anharmonic; anharmonic vibration

## 1. Introduction

### 1.1. Background

Specific heat at constant pressure ($C_p(T)$) is a thermodynamic quantity used by a wide range of academic fields of science and engineering. Unfortunately, $C_p(T)$ has historically been difficult to calculate. The quasi-harmonic approximation is indeed available, and for stiff materials it seems to work well for a broad range of temperatures, as in the case of ZrC [1,2]. However, for materials such as Ge, it has been shown that the anharmonic effects play a much larger role, and the quasi-harmonic approximation is not justified [3,4]. Without the quasi-harmonic approximation, $C_p(T)$ is much trickier to calculate. Indeed, one of the first things taught to young physicists is that $C_p(T)$ is easy to measure because it only requires the researcher to keep the force (pressure) constant but difficult to calculate because it requires the inclusion of the quasi-harmonic and anharmonic vibrational mode contributions. Conversely, specific heat at constant volume $C_v(T)$ is easy to calculate because it only requires the harmonic contributions, but is difficult to measure because the experimenter must keep the system's volume strictly constant. Work in this area has been limited to low-temperature, harmonic systems which satisfy the approximation $C_p \sim C_v$ [5], or the quasi-harmonic approximation which does not fully account for the anharmonic contributions properly—until recently [1,6]. Certainly, these various approximations are appropriate for stiff materials, such as diamond carbon [5] or rock salt ZrC [1,2,7], but are demonstrably inappropriate for more flexible (and therefore more anharmonic) materials, such as c-Ge [3,4].

### 1.2. Motivation/Purpose

For common materials many high-quality, consistent experiments have been conducted, and the values of $C_p(T)$ have been agreed upon for some time. However, for other common materials, such as wurtzite GaN, even when experimental literature is extensive, those experiments can be highly varied in their results [6]. In these situations, first-principles calculations are useful in determining which experiments are most likely to be the most accurate.

Furthermore, researchers do not always use common materials. Oftentimes, the system under study is a new material, such as a quasi-crystal [8] or a quantum dot [9,10], a complicated multi-component alloy [11], a carbon nanotube [12,13], or even one which is found in biology, such as spider silk proteins [14] or DNA [15]. For these types of materials, it is questionable whether experimental results are readily available; even if they are, there could be huge variability in the results, as was previously mentioned in the case of GaN [6]. This leaves the researcher to either perform the experiment herself, collaborate with an experimentalist, or complete the ab-initio calculations. Given the large investment of time and money that any experiment would require—this material would need to be fabricated and likely imaged before the experiment is even conducted—ab-initio simulations are a compelling option. Clearly, a method that allows for the quick, computationally efficient, reliable calculation of $C_p(T)$ for an arbitrary material, would be extremely valuable.

### 1.3. Status of the Field

Until recently [1,6,11], obtaining $C_p(T)$ from first principles was computationally prohibitive, as the third-order (or higher) vibrational modes were necessary [16]. Thankfully, both of the recent methods are computationally modest. However, the latter of the two developed by Duff and coworkers and termed the *two-stage upsampled thermodynamic integration using Langevin dynamics* (TU-TILD) uses finite-temperature density functional theory (FTDFT) for which, as of the time of this writing, no standardized codes exist. Therefore, FTDFT, and by extension TU-TILD, require considerable expertise to use effectively, not least of all because there is no known standardized way to develop reliable exchange-correlation functionals [17]. So while this method is very valuable and provides detailed information, particularly if one needs electron entropy (e.g., for a metal), TU-TILD is likely to be judged to be at least as difficult to implement as an experiment itself for many researchers.

The Beyond Quasi-Harmonic method, however, requires only standard ($T_{electron} = 0\,K$) density functional theory [6]. Specifically, we use the SIESTA code [18–20], but any of the many standardized codes available should work (VASP, Quantum Espresso, Abinit, etc.). The computational costs are modest, with the SiC system presented here representing 22 node hours on Stampede2, which, with parallel computing, can easily be brought down to 1–3 h of wall clock time.

In this paper, we calculate $C_p(T)$ for SiC and Ge, two common semiconducting materials which are lacking a set of reliable values in the literature for one or more temperature ranges. Our results for SiC are in excellent agreement with the available experiments we were able to find. We chose SiC because of its many applications—especially in semiconductor design, where overheating is a particular concern—and SiC is seen as a promising replacement to Si because of its good thermal properties [21,22]. Furthermore, we chose Ge because it is a well-known anharmonic material [3,4] that will provide a good demonstration of the ability of the Beyond Quasi-Harmonic method to account for anharmonic, rather than simply quasi-harmonic, effects.

### 1.4. Scope of the Paper/Summary of Paper

We organized the paper as follows: In Section 2, we describe the particular systems that we use in our example calculations, as well as the density functional theory (DFT) settings used to describe each atom involved. We then give a brief discussion of the Beyond Quasi-Harmonic method. In Section 3, we present data to determine what supercell size is best and acceptable for most materials given the nature of the anharmonic contributions we are trying to capture, and we present a set of recommended values for both the SiC and Ge materials. In Section 4, we summarize our findings and draw conclusions.

## 2. Methods

### 2.1. Density Functional Theory

We employ density functional theory for all our calculations; specifically, we use the SIESTA method [18–20]. The exchange–correlation potential is that of Ceperley–Alder [23], as parametrized by Perdew and Zunger [24] in the local density approximation. Norm-conserving pseudopotentials in the Kleinman–Bylander [25] form are used to remove the core electron orbitals from the calculations. We use double zeta basis sets for the C atoms and double zeta polarized (d-orbitals) for the Ge and Si atoms.

In order to study SiC and Ge, we built three supercells, one of which (216 Ge) is shown in Figure 1. The yellow atoms in Figure 1 are Ge atoms, which denote the hot block atoms, will be discussed in Section 2.2. Each of the three supercells is of the Zincblende structure, each of the SiC and Ge supercells we used for the calculations were 216 atoms, and a single 64-atom Ge supercell was used to determine the size of the effect on the system. We found the lattice constant for SiC to be 4.373 *A*, and 5.79 *A* for Ge. For both 216-atom supercells, we used a mesh cutoff of 250 Ry and a PAO energy shift of 0.2 eV, but a PAO energy shift of 0.25 eV for the 64-atom Ge supercell.

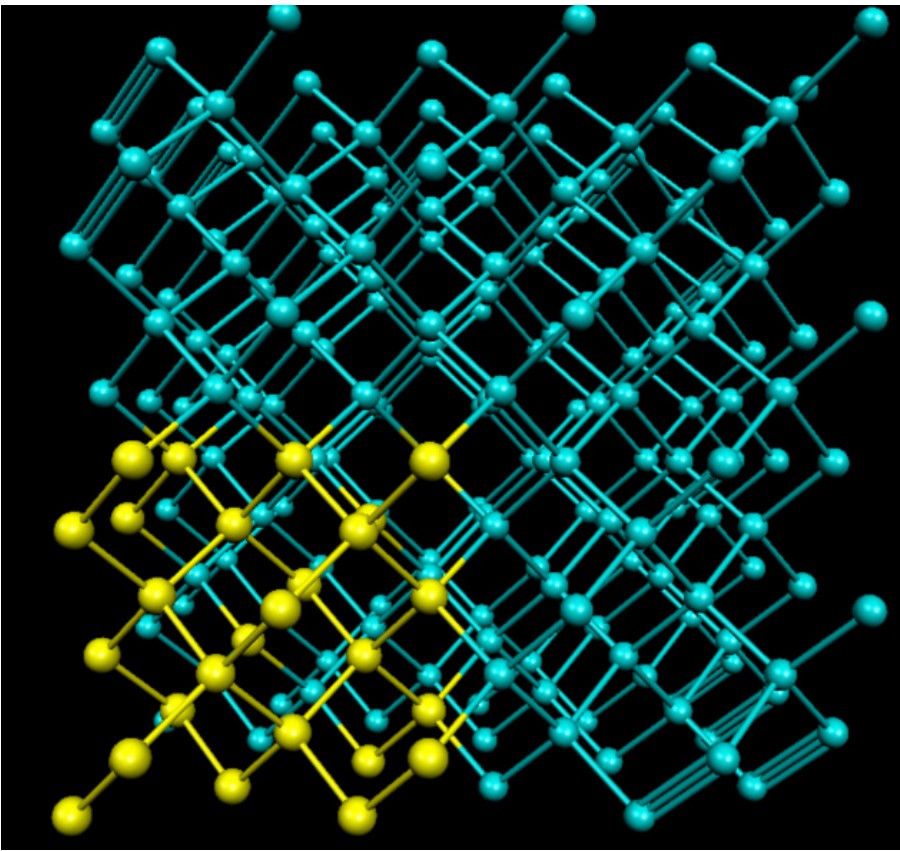

**Figure 1.** Shown above is an example of one of the supercells used in our calculations. The blue and yellow atoms are both Ge, where the yellow atoms denote the hot block atoms.

### 2.2. Beyond Quasi-Harmonic Method

The Beyond-Quasi Harmonic method has been described in detail previously [6], so it will only be summarized here. The central concept behind it is that supercell preparation is able to set up a system at a specific temperature without the use of a thermostat [26]. It is normally used to study heat flow [26,27]. The temperature of a region or system is defined by scaling the amplitudes of the normal vibrational modes by an amount determined by an inverse transform based on the Bose–Einstein distribution. However, supercell preparation only considers harmonic energy. So, the system is given a specific amount of

harmonic energy at a specific temperature, and then density functional theory is used to tell us—via the Kohn-Sham energy equations—how much energy the system really has, with the difference being the vibrational anharmonic energy of the system. However, it should be noted that these initial results from DFT cannot be used to calculate the enthalpy of the system because supercell preparation assumes that the average energy is $3NK_BT$, where $N$ is the number of atoms, $T$ is the temperature, and $K_B$ is the Boltzmann constant. Since this implicitly assumes equipartition, and therefore, that $C_v = 3NK_B = constant$, we lose all detail about $C_v(T)$. Therefore, the anharmonic energy is calculated at a set number of temperatures and is fit to a polynomial to find the anharmonic energy as a function of the temperature. Then, a derivative is taken, thus obtaining an equation for $C_p(T)$. However, when this is completed with an entire supercell, because of the repeated boundary conditions used, anharmonic energy is never seen, because the system's volume is kept strictly constant. For that reason, only $1/8$ of the supercell is heated up to hotter and hotter temperatures, but the overall harmonic temperature (and energy) of the supercell is kept at a constant 200 K. Implicitly, this assumes that there are no anharmonic effects at 200 K. This process is repeated many times (i.e. many microstates) for each temperature, and the results are averaged together.

## 3. Results and Discussion

Our results for the SiC system are shown in Figure 2. We can see that the data agree very well with our calculations. The polynomial that we would recommend using for the SiC system is given in two separate tables, which show the coefficients of a polynomial whose form is given in Equation (1), below. The coefficients are given in Tables 1 and 2 below, which are valid from 50 to 600 K and from 600 K to at least 1600 K, respectively.

$$C_p(T) = AT^5 + BT^4 + CT^3 + DT^2 + ET + F \tag{1}$$

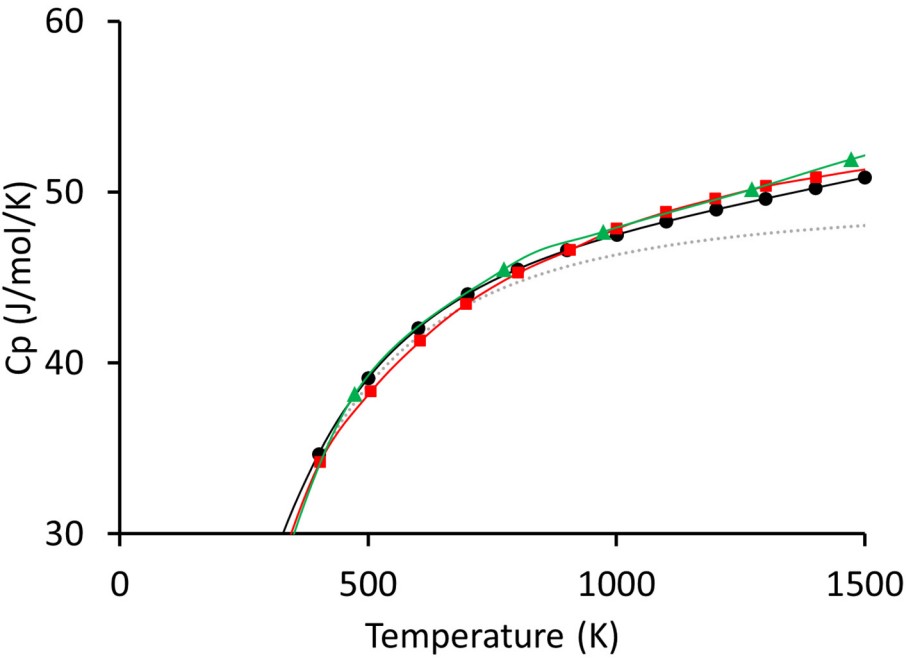

**Figure 2.** Shown above are our results compared to the experimental data available in the literature. The grey dotted line represents the calculated values of $C_v(T)$ (included for reference), the black circles are the $C_p(T)$ values from the present work, the red squares are from reference [28], and the green triangles are from reference [29].

**Table 1.** Coefficients for Equation (1) for the temperature range 50 K to 600 K, SiC (J/mol/k).

| | |
|---|---|
| A | $-2.812090475 \times 10^{-12}$ |
| B | $5.690248937 \times 10^{-9}$ |
| C | $-4.285321740 \times 10^{-6}$ |
| D | $1.344371869 \times 10^{-3}$ |
| E | $-6.020640219 \times 10^{-2}$ |
| F | $6.053784794 \times 10^{-1}$ |

**Table 2.** Coefficients for Equation (1) for the temperature range 600 K to 1600 K, SiC (J/mol/K).

| | |
|---|---|
| A | – |
| B | $-8.136069952 \times 10^{-12}$ |
| C | $4.358400273 \times 10^{-8}$ |
| D | $-9.051433696 \times 10^{-5}$ |
| E | $9.223260826 \times 10^{-2}$ |
| F | 10.52098061 |

Our results for both the 64-atom and 216-atom Ge supercell are shown in Figure 3. As pointed out by Nelin [3] and Leadbetter [4], the results of the quasi-harmonic results significantly differ from the generally accepted values. Conversely, the Beyond Quasi-Harmonic method is much better at reproducing the generally accepted experimental results as measured in reference [30] than the quasi-harmonic approximation. This indicates that 216 atoms are likely more than sufficient for the overwhelming majority of materials.

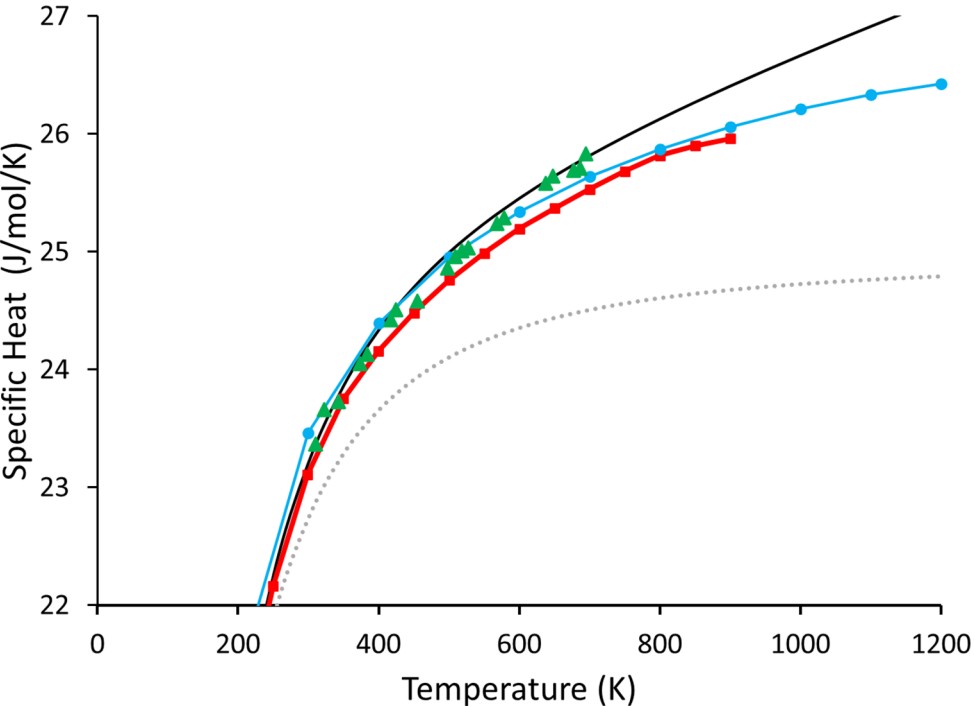

**Figure 3.** Shown above are our results for the 216- and 64-atom systems of Ge. The solid black line is the 216-atom cell from this work, and the blue circles are the 64-atom cell, also from this work. The grey dotted line represents the calculated values of $C_v(T)$ (included for reference). The green triangles are the values from reference [30], and the red squares are the quasi-harmonic values calculated in reference [3].

We can see the 64-atom cell is almost as good as the 216-atom cell for temperatures of up to about 600 K. However, as the temperature increases, it begins to resemble the inadequate values calculated in reference [3].

This is intimately linked to our rationale for believing the 64-atom cell does not adequately account for the anharmonic contributions. In the phonon density of states of the 64-atom cell, the normal modes are farther apart in frequency (and energy), and therefore, have a much lower probability of interacting. This relationship is seen mathematically from the interaction probability of two quantum particles (here, phonons; see chapter 3 of reference [16]): $P(i, f) = 2|\langle i|H'|f \rangle|^2 \frac{1 - \cos\left(E_f - E_i\right)t/\hbar}{\left(E_f - E_i\right)^2}$; where $i$ and $f$ are the initial and final states respectively, $E$ is energy, $t$ is time, $\hbar$ is the reduced Plank's constant, and $H'$ is the interaction Hamiltonian. This then means the increases in extra-harmonic energy above a certain temperature depend almost entirely on increases in volume rather than phonon interactions; this, in turn, means $C_p(T)$ can only depend on the volume changes and is then forced to match the quasi-harmonic approximation—exactly the behavior seen in Figure 3. This demonstrates that our method does, in fact, account for all the anharmonic vibrations.

Lastly, to our knowledge, there has not been a set of recommended values for Ge above 700 K. While there are some experiments [30–33], and some recommended values do exist up to 700 K [34], with one exception [33], we are unaware of anything that goes above 700 K. Furthermore, the results of reference [33] have a kink starting around 900 K, which we believe is likely a symptom of impurities on the sample or other difficulties arising from such a complicated experiment. For this reason, data from reference [33] was not considered when calculating our set of recommended values. However, for reference, all experimental data are shown in Figure 4. We give our recommended values for Ge from 300 K to 1200 K in Table 3 below:

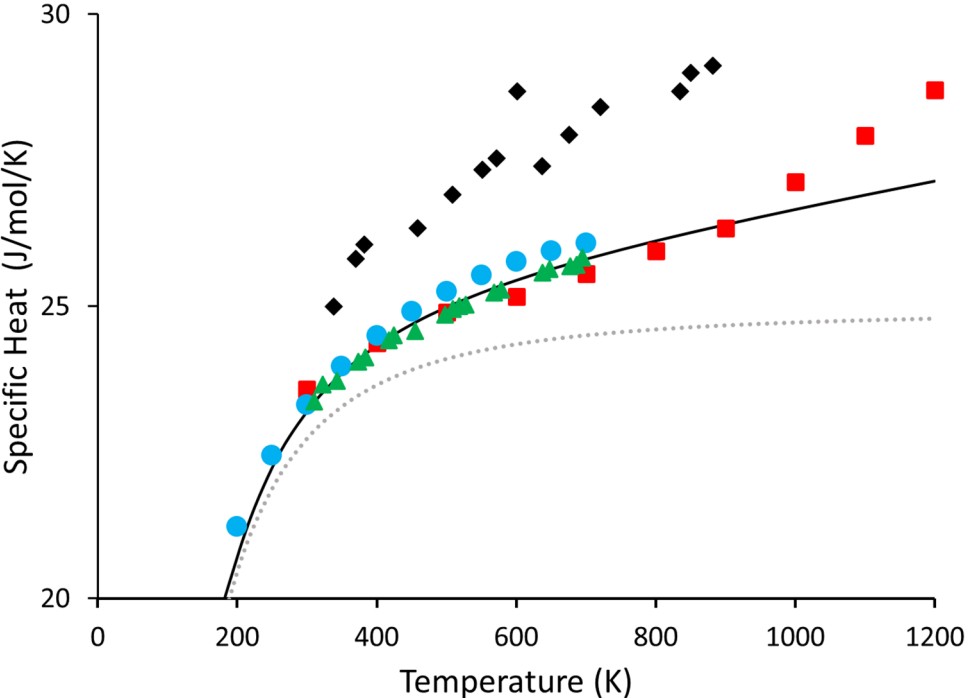

**Figure 4.** Shown above are all the experimental values of Ge that we are aware of. Blue circles are taken from [32], the red squares are taken from [33], the black diamonds are taken from reference [31], and the green triangles are the same as those in Figure 3, from [30]. The black line and the dotted grey line are the $C_p(T)$ and $C_v(T)$ from this work, respectively.

**Table 3.** Coefficients for Equation (1) for the temperature range 300 K to 1200 K for Ge (J/mol/K).

| | |
|---|---|
| $A$ | – |
| $B$ | $-1.238264848 \times 10^{-11}$ |
| $C$ | $4.420007261 \times 10^{-8}$ |
| $D$ | $5.902203596 \times 10^{-5}$ |
| $E$ | $3.773481439 \times 10^{-2}$ |
| $F$ | $16.13836107$ |

## 4. Conclusions

In conclusion, we demonstrated that the Beyond Quasi-Harmonic method takes into account all anharmonic vibrations of a dielectric material, by showing that we could exceed the results of the quasi-harmonic approximation in an anharmonic material, Ge. In addition, we also added a much-needed set of recommended values for the specific heat at constant pressure for SiC and Ge. From these values, one can easily calculate the vibrational enthalpy from the equation $E(T) = \int_{0K}^{T} C_p(T')dT'$, where $T$ is the temperature and $E$ is the enthalpy.

Future work will likely extend the BQH method to include other geometries, such as surfaces, nanotubes, and nanowires, among others.

**Funding:** This research received no external monetary funding, see acknowledgments.

**Institutional Review Board Statement:** Not applicable.

**Informed Consent Statement:** Not applicable.

**Data Availability Statement:** The data presented in this study are available on request from the corresponding author.

**Acknowledgments:** The authors would like to thank the Extreme Science and Engineering Discovery Environment (XSEDE), through allocation TG-PHY210056. XSEDE is supported by National Science Foundation grant number ACI-1548562. We'd also like to thank the Texas Advanced Computing Center (TACC) for generous amounts of computing time on Stempede2.

**Conflicts of Interest:** The authors declare no conflict of interest.

**Sample Availability:** Samples of the compounds are not available from the authors.

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
