# Peer review of "Vibrational Enthalpies of Solid Crystalline Materials"

_solids, doi:10.3390/solids3020023_

Round 1

Reviewer 1 Report

It is a well written manuscript I suggest to publish as it is. All of the conclusions is proven and the english is acceptable. The topic is not so important but it can be interesting for the experts. The quality of graph is reasonable. ALl of the work is present an acceptable form. ALl of the hyptheses is proven clearly. 

Author Response

We thank the referee for his or her comments and have changed our use of the first person with the passive voice throughout the methods section.

Reviewer 2 Report

Very nice work, which just contain minor flaws:

1) Title is a bit misleading. A better one would be: "Beyond Quasi-Harmonic method on SiC and Ge".

2)  The authors present an ab-initio calculated Cv, but not an ab-initio calculated Cp (Quasi-harmonic approximation using also an energy vs. volume plot etc.). Ref [3] seems to be a QHA on experimental data.

3) PZ-LDA and norm conserving pseudopotentials are used and I'm wondering why the authors didn't use the superior(?) PBE-GGA (+U) and PAW pseudopotentials instead

4) On some occasions the physical units like K, eV, Ry etc. are italic when they should be normal.

Author Response

1) Title is a bit misleading. A better one would be: "Beyond Quasi-Harmonic method on SiC and Ge".

In regards to this comment, we believe that title is appropriate to the findings of the paper, since the anharmonic vibrational energies are explicitly calculated, and can be obtained in general terms (J/mol, for example) by the simple equation E(T) = ∫Cp(T)dT.

In addition, our Cp is appropriate only for dielectric materials, no electrons are taken into account in this method, meaning that the energy from vibrations are the most important thing.  No Changes to the manuscript were done.

2)  The authors present an ab-initio calculated Cv, but not an ab-initio calculated Cp (Quasi-harmonic approximation using also an energy vs. volume plot etc.). Ref [3] seems to be a QHA on experimental data.

We respectfully disagree with the reviewer on this point.  We do calculate both Cp and Cv from first principles.  This is stated clearly in the Abstract, the method by which Cv and Cp are calculated is clearly detailed in ref [7], and the results are show in figures 2 and 4 for SiC and Ge respectively.  Comparison of our method to the QHA calculated in Ref[3] for Ge shows the superiority of our method to that of the QHA.  No Changes to the manuscript were done.

3) PZ-LDA and norm conserving pseudopotentials are used and I'm wondering why the authors didn't use the superior(?) PBE-GGA (+U) and PAW pseudopotentials instead.

The primary reason is because we know that the materials in question are well described by the LDA pseudopotentials (see especially ref. [6]), and we don't see any reason to fix what is not broken.  In addition, there is evidence that the LDA pseudopotentials are more accurate than the GGA, as can be seen in ref. [1]. No Changes to the manuscript were made.

4) On some occasions the physical units like K, eV, Ry etc. are italic when they should be normal.

We have corrected this wherever we found it, most notably at the of the second paragraph of the Methods section.

Reviewer 3 Report

The review of the manuscript: “Vibrational Enthalpies of Solid Crystalline Materials”

The manuscript contains 4 figures, 4 tables, with 34 references. 

Number of pages: 9

The author uses the same method as in 2019 in the reference 7 , only the choice of materials is new. 

The author argues that the Beyond Quasi Harmonic method takes into account all anharmonic vibrations of a dielectric material and gives better results than the quasi-harmonic approximation. 

The Beyond Quasi Harmonic is described previously in ref 7 and author doesn’t give 

more details about the method. The results are in good agreement with experimental results which are given in figures. Much-needed set of recommended values for the specific heat at constant pressure for SiC and Ge are results of the fit given in 3 tables.

The author didn’t explain why there are 2 tables for SiC in two different temperature regimes.  

The author uses the 216 atoms and once 64 atoms for the supercell, it would be very useful to know if a smaller number of atoms can provide the same result. 

The author uses the simplest functional LDA for the DFT, why not use the PBE functional  at least for structural optimisation? 

The results from quasi-harmonic approximation would give comparison of the methods a lot easier.

I do hope that formatting  will improve.

Author Response

The author didn’t explain why there are 2 tables for SiC in two different temperature regimes.

The reason for this is purely practical--combining the two temperature regions would require a ~11 term polynomial, which just seemed unwieldy. No Changes to the manuscript were made.

The author uses the 216 atoms and once 64 atoms for the supercell, it would be very useful to know if a smaller number of atoms can provide the same result.

As is shown in Figure 3 (Blue Circles), the 64 atom cell already under counts the anharmonic interactions, giving unreliable results.  So we do not see much value in another calculation for a 32 atom cell.  216 atoms, on the other hand gives reliable results for both the stiff SiC and anharmonic Ge, so we recommend 216 atom supercells.

The results from quasi-harmonic approximation would give comparison of the methods a lot easier.

This is a good point, but we already do this for Ge in figure3--The red squares are the quasi-harmonic approximation.  We show that the 64 atom cell becomes closer to this approximation at higher temperatures as the lack of phonon interactions becomes more apparent.  In this case there are fewer interactions because the 64 atom cell only has a very limited number of vibrational modes, which are farther apart in frequency.  Thus phonon interactions require interactions which are far less likely because of the greater difference in energy of the vibrational modes. We added clarification highlighted in blue to the manuscript for the previous two points that the referee brings up.

The author uses the simplest functional LDA for the DFT, why not use the PBE functional  at least for structural optimisation?

The primary reason is because  the LDA pseudopotentials are more accurate than the GGA, as can be seen in ref. [1]. No Changes to the manuscript were made.

Reviewer 4 Report

In this paper, the Beyond Quasi Harmonic method is proposed to analyze the specific heat of 3C-SiC and Ge, which has proved that the results are significantly better than those achieved by using the quasi-harmonic model when considering all anharmonic vibrations. The addressed topic is relevant and consistent with the scope of the journal. However, there are some points that the authors have to consider before publication.

  1. In this paper, the title is 'Vibrational Enthalpies of Solid Crystalline Materials', but the specific heat Cp(T) is mentioned in the abstract part and other parts. Please revise the title of the manuscript to clarify the research topic or provide some necessary explanations in the abstract.
  2. In Section 1.3, there are a few works of literature to describe the corresponding research status. Please supplement some ones that investigate the specific heat parameters for SiC and Ge materials.
  3. In Section 2, please provide the necessary calculation, equation derivation and detailed descriptions of key variables.
  4. In Section 3, the test setup is unclear. Please describe how the experimental data is obtained.
  5. The authors are advised to write more in the passive voice.
  6. Some references are incorrectly formatted. Please revise them.

Author Response

1. In this paper, the title is 'Vibrational Enthalpies of Solid Crystalline Materials', but the specific heat Cp(T) is mentioned in the abstract part and other parts. Please revise the title of the manuscript to clarify the research topic or provide some necessary explanations in the abstract.

In regards to this comment, we believe that title is at most merely a trivial step removed from the findings of the paper, since the anharmonic vibrational energies are explicitly calculated, and can be obtained in general terms (J/mol, for example) by the simple equation E(T) = ∫Cp(T)dT'.  We added this point to the conclusions section of the manuscript, highlighted in blue.

2. In Section 1.3, there are a few works of literature to describe the corresponding research status. Please supplement some ones that investigate the specific heat parameters for SiC and Ge materials.

The citations for Ge are references 3, 4, 28 - 34.  The citations for SiC are given in 28 and 29.  Ref. 34 does also give values for SiC, but these values are wrong (negative values are obtained for Cp), and so are not included.  We are unaware of any other citations which would be relevant to our work.  If the referee would like to point some out, and we agree that they are relevant, we would be happy to add them.   

3. In Section 2, please provide the necessary calculation, equation derivation and detailed descriptions of key variables.

We do this by reference.  They are already included in ref. [7], we do not include them here because they are not the focus of research in this publication. No Changes made to the manuscript.

4. In Section 3, the test setup is unclear. Please describe how the experimental data is obtained.

There is not an experimental component to this work, all experimental data are obtained from the works cited in the paper. No Changes made to the manuscript.

5. The authors are advised to write more in the passive voice.

Agreed.  The passive voice is much more appropriate, particularly in the methods section.  Changes for this purpose are highlighted in yellow.

Round 2

Reviewer 4 Report

The revised manuscript replies and revisions are reasonable.